# The Role of Nitrogen-Doped TiO_2_ Supported by Platinum Catalyst Synthesized via Various Mode Preparations for Photocatalytic Enhancement

**DOI:** 10.3390/nano12223998

**Published:** 2022-11-13

**Authors:** Nadiah Sabihah Natar, Nureel Imanina Abdul Ghani, Siti Raihan Hamzah, Muhammad Afiq Rosli, Nur Aien Muhamad, Mohammad Saifulddin Azami, Mohd Azlan Mohd Ishak, Sharin Razak, Wan Izhan Nawawi

**Affiliations:** 1Faculty of Applied Sciences, Universiti Teknologi MARA Cawangan Perlis, Arau 02600, Perlis, Malaysia; 2Faculty of Applied Sciences, Universiti Teknologi MARA, Shah Alam 40450, Selangor, Malaysia; 3Faculty of Chemical Engineering Technology, Universiti Malaysia Perlis, Arau 02600, Perlis, Malaysia

**Keywords:** nitrogen-titanium dioxide, platinum nanoparticles, oxygen vacancy, chemical oxygen demand, photocatalysis

## Abstract

The limitations of TiO_2_ as a photocatalyst such as the larger bandgap energy, which only activates under the UV region, give a lower photocatalytic activity. This study reports the role of the N and Pt co-dopant on the modification of the TiO_2_ photocatalyst for photocatalytic degradation of methylene blue dye under different mode preparations, i.e., sequential and vice-versa modes. The sequential mode preparation of the N and Pt co-dopant TiO_2_ photocatalyst consisted of the initial preparation of the N-doped TiO_2_ (N-TiO_2_) under the calcination method, which was then further doped with platinum (Pt) through the photodeposition process labeled as NP_seq_-TiO_2_, while the vice-versa mode was labeled as PN_rev_-TiO_2_. About 1.58 wt.% of N element was found in the NP_seq_-TiO_2_ photocatalyst, while there was no presence of N element detected in PN_rev_-TiO_2_, confirmed through an elemental analyzer (CHNS-O) and (EDX) analysis. The optimum weight percentage of Pt for both modes was detected at about ±2.0 wt.%, which was confirmed by inductively coupled plasma-emission spectroscopy (ICP-OES). The photoactivity under methylene blue (MB) dye degradation of the NP_seq_-TiO_2_ photocatalyst was 2 and 1.5 times faster compared to the unmodified TiO_2_ and PN_rev_-TiO_2_, where the photodegradation rates were, ca., 0.065 min^−1^ and 0.078 min^−1^, respectively. This was due to the N elements being incorporated with the TiO_2_ lattice, which was proven by UV-Vis/DRS where the bandgap energy of NP_seq_-TiO_2_ was reduced from 3.2 eV to 2.9 eV. In addition, the N generated a stronger PL signal due to the formation of oxygen vacancies defects on the surface of the NP_seq_-TiO_2_ photocatalyst. The higher specific surface area as well as higher pore volume for the NP_seq_-TiO_2_ photocatalyst enhanced its photocatalytic activity. Moreover, the NP_seq_-TiO_2_ showed the lowest COD value, and it was completely mineralized after 7 h of light irradiation. The preparation order did not affect the Pt dopant but did for the N element. Therefore, it is significant to investigate different mode preparations of the N and Pt co-dopant for the modification of TiO_2_ to produce a good-quality photocatalyst for photocatalytic study under the photodegradation of MB dye.

## 1. Introduction

Heterogenous photocatalysis has broadly been studied for environmental purposes as it one of the good alternative processes for wastewater treatment. It has been developed since Fujishima and Honda discovered the photocatalytic water-splitting using titanium dioxide as an electrode [1]. Most of the heterogenous photocatalysis has been conducted based on the semiconductor material from metal oxide elements. Among the available semiconductors, titanium dioxide (TiO_2_) [2], zinc oxide (ZnO) [3], ZnS [4], and WO_3_ [5] have bandgap energies sufficient to activate the chemical reactions and exhibit a good efficiency on the degradation of organic pollutants upon its absorption of light energy. However, this research focuses on the TiO_2_ semiconductor for photocatalytic enhancement due to its advantages as an environmentally friendly material, low-cost chemical with high photocatalytic activity, and abundant material [6,7]. The crystal structure of TiO_2_ is usually made up of anatase, rutile, and brookite phases. The anatase phase has an indirect bandgap energy of about 3.2 eV, while the rutile phase has a direct bandgap energy of about 3.0 eV. Even though the rutile phase has a lower bandgap energy, it gives a rapid electron–hole recombination; thus, it has a decreased photocatalytic performance [8]. It was, thus, proposed to apply the combination phase for the TiO_2_ photocatalyst to a ratio of 80:20 anatase to rutile phase for photocatalytic enhancement. Apart from the photocatalysis process, TiO_2_ has also been applied in hydrogen production [9], dye-synthesized solar cells (DSSCs) [10], sterilization [11], defoggers [12], self-cleaning [13], and air purification [14,15].

Due to the larger bandgap of the TiO_2_ semiconductor, TiO_2_ needs ultraviolet (UV) light irradiation to generate the electron-pair. As shown in Figure 1, the photogenerated electron (e^−^) at the VB was excited to the CB, allowing the oxidation process and, thus, forming a superoxide radical, while the formation of a hole (h^+^) at the VB will undergo a reduction process, forming a hydroxyl radical. The formation of these radicals is important for the photocatalysis process where it will degrade the harmful pollutants in the wastewater [16]. Hence, the TiO_2_ suffers from an inefficient utilization of visible-light irradiation and a rapid rate of electron–hole pair recombination [17,18,19]. Thus, the modification of TiO_2_ by doping with a non-metal group such as nitrogen (N), sulfur (S), and carbon (C) elements and a metal group such as platinum (Pt), aurum (Au), and silver (Ag) was carried out as it was a good alternative to overcome the disadvantages of the TiO_2_ photocatalyst. According to Serpone et al. [20], the oxygen defect in the TiO_2_ lattice, which will then be substituted with the non-metal elements, has potential to introduce a new bandgap energy below the valence band known as the sub-band. Thus, the modified TiO_2_ photocatalyst will generate a higher photocatalytic activity as it has reduced the bandgap energy, which will then increase the energy wavelength toward visible-light irradiation (λ > 400 nm) [21,22,23]. Among the elements in the non-metal group, the N element is the most preferable for modification of TiO_2_ due to its potential to introduce a new bandgap energy below the valence band; it is also capable of adjusting the optical bandgap and acting as a superficial donor [24,25,26]. However, it is easy for the excited electron toward the sub-band to recombine with the hole in the valence band; thus, it will decrease the photocatalytic performance. Hence, noble metal has been proven to reduce the rate of electron–hole recombination by scavenging the excited electron at the conduction band and further undergoing the oxidation process for degradation of organic pollutants [27]. According to Kumaravel et al. [28], the metal dopants on the TiO_2_ photocatalyst can increase the efficiency of H_2_ production through Schottky barrier formation and surface plasmon resonance (SPR). The formation of a Schottky barrier at the TiO_2_-metal surface will be capable of preventing the electron–hole recombination. Among the elements in the noble metal, the Pt dopant is the most suitable element to be applied in the modification of the TiO_2_ photocatalyst. As has been reported by Trejo-Tzab et al. [29], the Pt dopant forms the highest Schottky barrier as compared to the Ag and Au dopants; thus, it will support the excitation electrons capture and decrease the electron–hole pair recombination rate.

As a result of the sample preparation method, many studies have reported that changes in the morphology and surface defect of TiO_2_ will affect the photoactivity. Chen et al. [30] reported that the formation of Ti^3+^ self-doped TiO_2_ hollow nanocrystals synthesized through the microwave-assisted ionic liquid solvothermal method followed by the vacuum-activation method enhanced the photocatalytic performance. This was due to the formation of oxygen vacancies whereby the synthesized sample gave an intense EPR signal at g = 1.97–2.02. In addition, the mesoporous structure was obtained as an ionic liquid [Bmim][BF_4_] involved in the control of the morphological agent through a dissolution-recrystallisation process. Moreover, Serga et al. [31] developed a new method of producing nanocrystalline TiO_2_ powder using the extraction–pyrolytic method (EPM). It was observed that the heating temperature influenced the phase composition and the average particle size of the synthesized photocatalyst. Thus, it is necessary to study the optimal temperature on the sample preparation, where it will determine the photocatalytic performance. In this research, the innovation of modification of the TiO_2_ photocatalyst by combining the N and Pt elements is significant to study its sequences of mode preparation. This research provides an important contribution on two possible methods: sequential or vice-versa mode preparation, to synthesize the commercially available TiO_2_ with N (non-metal) and Pt (noble metal) elements, which have a great influence on the percentage of N content. Moreover, the synergistic effect between N and Pt that affects the photocatalytic activity is also reported. Thus, this study is important for future reference.

## 2. Materials and Methodology

### 2.1. Synthesis of Photocatalyst

The commercially available Degussa P25 (TiO_2_) was used as initial raw material where the percent ratio of the anatase and rutile phase was 80:20. Urea (CH₄N₂O) was used as an N precursor supplied by R&M Chemicals and underwent the calcination process under an N_2_ gas condition for 2 h at a temperature 450 °C according to the method reported by Natar et al. [32]. A yellowish powder was obtained, which denoted the nitrogen-doped TiO_2_ (N-TiO_2_) photocatalyst. Then, about 0.50 g of N-TiO_2_ photocatalyst was mixed with 2 wt.% hexachloroplatinic (IV) acid hexahydrate (H_2_PtCl_6_.H_2_O, Sigma-Aldrich, St. Louis, MO, USA) as a platinum precursor, isopropyl alcohol (C_3_H_8_O, Systerm Chemicals) as a sacrificial agent, and distilled water. The solution was stirred and irradiated for an hour under a 250-Watt high-intensity lamp (visible-light irradiance of 2889 W m^−2^ and UV-light irradiance leakage of 436 W m^−2^) for the photodeposition process. A brown-greyish solution was obtained, which was further centrifuged, washed, and dried overnight at 100 °C. The brown-greyish powder was labeled as platinum, nitrogen co-doped TiO_2_ (NP_seq_-TiO_2_). The calcination treatment followed by the photodeposition process was devoted to sequential mode preparation while the vice-versa of this mode preparation was labeled as the PN_ver_-TiO_2_ photocatalyst. The color for the NP_seq_-TiO_2_ and PN_ver_-TiO_2_ photocatalysts was observed as a brown-greyish powder and greenish powder, respectively. The mode preparations of NP_seq_-TiO_2_ and PN_rev_-TiO_2_ were well constructed through appropriate methods, as shown in Figure 2.

### 2.2. Photocatalytic Activity

For the photocatalytic activity study, about 12 mg L^−1^ of methylene blue (MB) dye was used as an organic pollutant model. The volume of MB dye was measured for 20 mL and sonicated for a minute with 0.03 g of photocatalysts to obtain a homogenized solution. The sonicated solution was poured into a 15 cm × 10 cm × 8 cm (L × B × H) glass cell and irradiated for an hour using a 55-Watt fluorescent lamp (visible-light irradiance of 187 W m^−2^ and UV-light irradiance leakage of 3.15 W m^−2^). The adsorption study was conducted for 30 min to achieve an equilibrium rate before performing the photocatalytic study experiment. The aeration of oxygen was constantly supplied by an NS 7200 aquarium pump. The degradation of MB dye was determined by taking 3 mL of the treated solution in every 15 min interval and filtering it using a 22 µm nylon syringe filter (HmbG Chemicals) prior to analysis of the sample. The absorption was detected by using a HACH DR 1900 spectrophotometer at a wavelength of 661 nm. The photocatalytic efficiency was calculated based on Equation (1) [33]:(1)$$\mathrm{MB}\;\mathrm{dye}\;\mathrm{degradation}\;(\%)=\frac{C_{o}-C_{t}}{C_{o}}\times 100$$
where *C_o_* and *C_t_* are the initial and remaining concentration of the MB in solution, respectively, and *C_t_* is the absorbance at any irradiation of time. Meanwhile, the kinetics of the photocatalytic degradation of MB dye was calculated based on the Langmuir–Hinshelwood pseudo-first-order kinetic model as stated in Equation (2). The pseudo-first-order rate constant, *k_app_* (min^−1^), was calculated from the slope of In (*C_o_*/*C_t_*) versus irradiation time, *t*. The same procedure was repeated by replacing with unmodified TiO_2_ and PN_rev_-TiO_2_ photocatalysts.
(2)$$In\;\frac{C_{o}}{C_{t}}=k_{app}t$$

### 2.3. Characterization

The synthesized sample was digested with the microwave digestion method using a 3:1 ratio of aqua regia for 2 h prior to analysis by the inductively coupled plasma-optical emission spectrometer (ICP-OES, PerkinElmer, Waltham, MA, USA), to determine the Pt composition in the prepared sample. The elemental analyzer (CHNS-O, Thermo Scientific Flash 2000 Organic Elemental Analyzer, Cambridge, UK) was used to verify the content of N element in the NP_seq_-TiO_2_ and PN_versa_-TiO_2_ photocatalysts. A Field Emission-Scanning Electron Microscope (FE-SEM, Zeiss SUPRA 55-VP, Oberkochen, Germany) was used to study the morphologies of the photocatalyst while the EDX and mapping analysis was performed to determine the percentage of element and to observe the distribution composition of the prepared sample. High-resolution transmission electron microscopy (HR-TEM, Jeol-JEM-2100F, Tokyo, Japan) was conducted to study the surface structure, average particle size, as well as the distribution of Pt element in the prepared photocatalyst. The X-ray diffraction (XRD, PANalytical X’pert PRO, model PW 3040/60, Malvern, UK) patterns of the prepared sample were analyzed at a scan rate of 10 °C (2θ)/min with a Cu Kα X-ray source and wavelength of λ = 1.5415 Å. Furthermore, the ratio for preparation of the KBr pellet was fixed at 1:80 and pressed at 10 ton (t) for a minute. Then, the KBr pellet was analyzed using a Fourier transform infrared transmittance spectrometer (FTIR, PerkinElmer FT-IR Spectrometer Frontier, Waltham, MA, USA) to identify the bonding presence in the prepared sample. The surface characteristics of the sample were evaluated through Brunauer–Emmett-Teller (BET) analysis. The nitrogen adsorption–desorption method was used to estimate the particle surface area, pore diameter, and pore volume carried out using the model AUTOSORB-1 Series (Quantachrome Instruments, Boynton Beach, FL, USA) at a temperature of 200 °C. The BET plot was measured at a diameter less than 2481.5 Å at P/P_0_ = 0.99220 and calculated using the BJH method. The optical properties of the photocatalysts were investigated using a UV-Vis diffuse reflectance spectrometer (UV-Vis/DRS, PerkinElmer LAMBDA 35, Waltham, MA, USA) and photoluminescence analysis (PL, Photon System Deep UV (DUV) Mini PL/RAMAN Spectrometer, Covina, CA, USA) to study the fate of electron–hole recombination at an excitation wavelength 325 nm. Next, the chemical oxygen demand analysis (COD) was conducted to obtain the details of the photocatalytic degradation of MB dye using a HACH DR 1900 spectrophotometer. About 2 mL of the sample was taken and mixed with chemical oxygen demand digestion solution (HACH, Malaysia) and then digested for 2 h at 150 °C by using the COD digester, HACH DRB 200, prior to the COD reading being taken. The COD graph is plotted and discussed in Section 3.7 below.

## 3. Results and Discussion

### 3.1. Crystallinity

The crystallinity of the synthesized NP_seq_-TiO_2_ and PN_rev_-TiO_2_ samples was investigated by XRD analysis, as shown in Figure 3. The diffraction peak of the synthesized sample was similar to that of the unmodified TiO_2_ photocatalyst (JCPDS No. 01-075-1537). The anatase diffraction peaks occurring at 2θ values at 25.6°, 36.3°, 38.8°, 48.3°, 55.2°, 64.3°, 69.1°, 70.5°, 76.3°, and 82.7° corresponded to d_101_, d_103_, d_004_, d_200_, d_211_, d_204_, d_116_, d_220_, d_301_, and d_303_. No shifted peak was observed for the NP_seq_-TiO_2_ and PN_rev_-TiO_2_ samples upon adding the N and Pt precursor on TiO_2_. Small additional peaks were at 39.5°, 44.8°, and 64.5° for the NP_seq_-TiO_2_ and PN_rev_-TiO_2_ photocatalysts, which represented the Pt peak [34,35,36]. This confirmed the presence of Pt in the NP_seq_-TiO_2_ and PN_rev_-TiO_2_ photocatalysts. Furthermore, a lower intensity of the rutile phase was detected at 27.7°, 41.5°, and 44.5° corresponding to d_110_, d_111_, and d_210_. However, there was no phase transformation, due to the lower temperature that was used in the calcination treatment. Lin et al. [37] reported that the transformation from the anatase to rutile phase requires a calcination temperature greater than 700 °C. Table 1 shows the structural properties of the TiO_2_, NP_seq_-TiO_2_, and PN_rev_-TiO_2_ photocatalysts. The crystallinity index, CI, for all the prepared photocatalysts was calculated using Equation (3), as shown below. The unmodified TiO_2_ showed the highest crystallinity index, ca., 81.33%, as compared to the NP_seq_-TiO_2_ and PN_rev_-TiO_2_ photocatalysts, ca., 80.81% and 79.99%, respectively. The decrement in crystallinity for NP_seq_-TiO_2_ and PN_rev_-TiO_2_ photocatalysts was due to the presence of N and Pt bonding incorporated with the TiO_2_ photocatalyst. However, Chaturvedi et al. [38] reported that the heat treatment may influence the crystallinity of the anatase phase as well as the crystallite size of the photocatalyst. The average crystallite size, D, of the prepared sample was calculated using Scherer’s equation, as shown below in Equation (4), and it was found to be in the range of 32–41 nm for all the prepared sample. As calculated using Equation (5), the d-spacing for all the prepared sample was, ca., 0.35 nm for the lattice (101), which was represented as an anatase dominant peak.
(3)$$\mathrm{Crystallinity}\;\mathrm{Index}\;(\mathrm{CI},\;\%)=\frac{\mathrm{Area}\;\mathrm{of}\;\mathrm{crystalline}\;\mathrm{peaks}}{\mathrm{Area}\;\mathrm{of}\;\mathrm{all}\;\mathrm{peaks}\;\left(\mathrm{crystalline}\;+\;\mathrm{amorphous}\right)}\times 100\;$$
(4)$$\mathrm{Crystallite}\;\mathrm{size}\;(D,\;\mathrm{nm})=\frac{K\lambda }{\beta \cos\theta }\;$$
where K = 0.9 (Scherrer’ s constant) and β is the FWHM value:(5)$$\text{d-spacing}\;(d,\;\mathrm{nm})=\frac{n\lambda }{2\sin\theta }\;\mathrm{where}\;\lambda =0.15406\;n$$

### 3.2. Fourier Transform Infrared Spectroscopy (FTIR) Analysis

The FTIR spectra ranging from 400 cm^−1^ to 4000 cm^−1^ wavenumbers for the TiO_2_, NP_seq_-TiO_2_, and PN_rev_-TiO_2_ photocatalysts are shown in Figure 4. No additional vibration peak was observed for all synthesized samples. The peak at 615 cm^−1^ and below represented a strong stretching and vibration of Ti-O and Ti-O-Ti bonds [39], while the peaks at 1400 cm^−1^, 1624 cm^−1^, and 3147 cm^−1^ corresponded to the O-H stretching and vibration bonding. The NP_seq_-TiO_2_ photocatalyst showed a higher intensity of O-H stretching as compared to the PN_rev_-TiO_2_ and unmodified TiO_2_ photocatalyst. This was due to the NP_seq_-TiO_2_ photocatalyst having a higher surface-adsorbed water and presence of hydroxyl molecules. The higher formation of hydroxyl radicals will give a rapid photodegradation of methylene blue dye by forming a reactive species, hydroxyl radicals [40]. In addition, the surface of hydroxyl radicals may act as an absorption center for O_2_ molecules, which then forms a hydroxyl radical to increase the photocatalytic activity [41].

### 3.3. Morphological and Structural Analysis

As shown in Figure 5, the FE-SEM with EDX analysis was performed to observe the morphological structure and verify the elemental compositions such as Ti, O, N, and Pt. In Figure 5a, the agglomeration of spherical particle size was observed for unmodified TiO_2_, while the morphology of the NP_seq_-TiO_2_ photocatalyst formed an irregular shape of particles and no clump particles as compared to the morphology in the PN_rev_-TiO_2_ photocatalyst where it contributed a larger cluster of particles, as seen in Figure 5c,e. Meanwhile, the EDX analysis found that the composition of the elements in the TiO_2_ photocatalyst did not change significantly, because the commercially available TiO_2_ was used as the starting material. As a result, the percentage of Ti and O was found to be approximately 46.8 wt.% and 53.2 wt.%, respectively (see Figure 5b), where the weight percent is comparable with a previous study [42]. Furthermore, the main concern for this research was the presence of N in the NP_seq_-TiO_2_ and PN_rev_-TiO_2_ photocatalysts. Based on the EDX analysis, the N content was successfully detected in the NP_seq_-TiO_2_ photocatalyst, ca., 0.6 wt.%, while no presence of N elements was detected for the PN_rev_-TiO_2_ photocatalyst, as shown in Figure 5d,f, respectively. This might be due to the low interaction bonding between the N and Pt-TiO_2_ photocatalyst prepared via calcination treatment. The weak bonding caused the N to be eliminated more easily during the photodegradation process, which affected the photocatalytic performance of the PN_rev_-TiO_2_ photocatalyst. However, there was no significant change in the availability of Pt in NP_seq_-TiO_2_ and PN_rev_-TiO_2_ photocatalysts.

Figure 6 shows the HR-TEM image, average particle size distribution, and mapping analysis for (a) NP_seq_-TiO_2_ and (b) PN_rev_-TiO_2_ photocatalysts. As shown in Figure 6a, the lattice fringes were obtained at about 0.35 nm for anatase TiO_2_ at plane (101), while in Figure 6b, the lattice fringe for Pt nanoparticles was obtained at about 0.19 nm and 0.23 nm, which was attributed to the plane (111) [43]. The plotted histogram for the average particle size distribution was calculated using ImageJ software. The particle size for NP_seq_-TiO_2_ and PN_rev_-TiO_2_ photocatalysts was obtained at about 19 nm and 24 nm, respectively. In addition, the Pt nanoparticles showed a uniform directional distribution for both mode preparations as it can be seen through mapping analysis. Hence, the HR-TEM images confirmed that the Pt nanoparticles were successfully being deposited on the TiO_2_ surface either via the sequential or vice-versa method.

The nitrogen adsorption–desorption isotherms for TiO_2_, NP_seq_-TiO_2_, and PN_rev_-TiO_2_ photocatalysts are shown in Figure 7. All the samples exhibited a type IV isotherm and revealed a mesoporous structure. The BET specific surface area can be defined as the exposed surface area of the nanoparticle per unit mass. It is one of the important factors to determine excellent photocatalyst characteristics on photocatalytic activity. According to Song et al. [44], the BET surface area, pore volume, and average pore diameter are inversely proportional to the particle size of nanoparticles, which will affect the photocatalytic performance. As a result, the NP_seq_-TiO_2_ photocatalyst showed a higher BET surface area, ca., 57 m^2^ g^−1^, as well as an increase in pore volume and pore diameter compared to the TiO_2_ and PN_rev_-TiO_2_ photocatalysts, 54 m^2^ g^−1^ and 25 m^2^ g^−1^, respectively (see Table 1). By assuming that the particle size of TiO_2_ has the same spherical shape and size, the average particle size, D, for TiO_2_ was estimated using Equation (6) as follows [45].
D = 6000/(S_BET_ × p)(6)
where S_BET_ is the specific surface area obtained from BET analysis and p is the true density for titania (p = 4.2 g/mL). However, the average particle size for NP_seq_-TiO_2_ and PN_rev_-TiO_2_ photocatalysts cannot be calculated using this equation, due to its irregular morphology structure that was observed through the FE-SEM image. Based on the HR-TEM image, the NP_seq_-TiO_2_ had the smallest particle size, ca., 19 nm as compared with the PN_rev_-TiO_2_ photocatalyst, which was obtained at about 26 nm. Thus, the smaller size of the NP_seq_-TiO_2_ nanoparticle produced a good photocatalytic activity as it increased the amount of surface adsorbed on the active site area.

### 3.4. Elemental Analysis (CHNS-O) and (ICP/OES)

The CHNS analyzer was used to provide secondary information on organic compounds such as carbon, hydrogen, nitrogen, sulfur, and oxygen contained in the sample. The elemental analysis was carried out in order to confirm the presence of N in the prepared sample. Table 2 shows a sample preparation and the pseudo-first-order rate constant, k, for the photocatalytic degradation of 12 mg L^−1^ of MB dye for TiO_2_, NP_seq_-TiO_2_, and PN_rev_-TiO_2_ photocatalysts irradiated with a 55-Watt fluorescent lamp for 60 min. As shown in Table 2, the nitrogen was successfully doped with TiO_2_ where about 0.60 wt.% and 1.58 wt.% N were detected in the NP_seq_-TiO_2_ photocatalyst, which was analyzed by EDX and CHNS-O analysis, respectively. According to Samsudin et al. [46], the difference in nitrogen content in the CHNS-O analyzer and EDX analysis was because the CHNS-O analyzer measured the total bulk nitrogen while EDX measured the nitrogen content on the surface of the photocatalyst. However, the PN_rev_-TiO_2_ photocatalyst showed a zero-weight percent of N, which was analyzed through CHNS-O and EDX analysis. Therefore, the availability of N in TiO_2_ modification was identified as prepared via sequential mode preparation. Based on the result, the sequence of the calcination and photodeposition process is important for investigation of the combination of N and Pt co-doped with commercially available TiO_2_. Furthermore, the concentration of the Pt in the prepared sample was, ca., 2.10 wt.% and 1.70 wt.% for NP_seq_-TiO_2_ and PN_rev_-TiO_2_ photocatalysts analyzed by EDX analysis, respectively. To further confirm the Pt content, the ICP-OES analysis was conducted and about 2.10 wt.% and 2.20 wt.% were obtained for NP_seq_-TiO_2_ and PN_ver_-TiO_2_ photocatalysts, respectively. Thus, the preparation of photocatalysts by the sequential and vice-versa modes did not have a significant effect on Pt doping but did have a great influence on the N concentration.

### 3.5. Optical Studies

Figure 8 shows the UV-Visible diffuse reflectance spectra (UV-Vis/DRS) of TiO_2_, NP_seq_-TiO_2_, and PN_rev_-TiO_2_ photocatalysts for (a) absorbance spectra and (b) Tauc’s plot. The wavelength was plotted ranging from 350 to 550 nm. The absorption edge in Figure 8a was around 400 nm for the unmodified TiO_2_ and PN_rev_-TiO_2_ photocatalysts where it exhibited a light absorbance in the UV region (200–400 nm). Moreover, the absorption edge for NP_seq_-TiO_2_ was redshifted toward the visible-light region, ca., 420 nm, upon light absorption. It explained the presence of N forming a new energy level above the valence band, thus reducing the bandgap energy of the photocatalyst [47,48]. Figure 8b shows the extrapolated graph to determine the energy bandgap of the unmodified TiO_2_, NP_seq_-TiO_2_, and PN_rev_-TiO_2_ photocatalysts by applying Tauc’s plot and Equations (7) and (8) [49].
(α*h*ν) = (*h*ν − E_g_)*^n^*(7)
(F(R)*h*ν)^1/*n*^ = (*h*ν − E_g_)(8)
where α is the absorption coefficient, *h* is Planck’s constant, ν is the frequency of incident light, 1/*n* is the indirect bandgap energy, and E_g_ is the bandgap energy. The estimation of the bandgap energy was carried out by extrapolating the linear portion curve to the zero value of the *y*-axis. From Tauc’s plot, the bandgap energy for unmodified TiO_2_ was, ca., 3.2 eV, while the bandgap energies for the NP_seq_-TiO_2_ and PN_rev_-TiO_2_ photocatalysts were, ca., 2.9 eV and 3.1 eV, respectively. There were no significant changes between the bandgap energy of unmodified TiO_2_ and PN_rev_-TiO_2_ photocatalysts, due the absence of the N element in the PN_rev_-TiO_2_ sample, which was supported by EDX (see Figure 5f) and CHNS-O analysis (see Table 2). The sequential mode preparation for the NP_seq_-TiO_2_ photocatalyst is significant for study where it confirmed the availability of N in the prepared sample. Hence, the reduction in the bandgap energy and extension of light absorption in the NP_seq_-TiO_2_ photocatalyst were due to the presence of N dopant [50].

Photoluminescence (PL) analysis has usually been studied to obtain information regarding the efficiency of charge separation, carrier trapping, charge transfer, and surface defects on photocatalysts. It also explains the behavior of the electron–hole pairs in semiconductors [51,52]. Figure 8c shows the PL emission spectrum for unmodified TiO_2_, NP_seq_-TiO_2_, and PN_rev_-TiO_2_ photocatalysts. As we can see, the NP_seq_-TiO_2_ photocatalyst had two obvious peaks at about 444 nm indicated as a band-to-band PL signal and 579 nm represented as a band to sub-band PL signal. Based on the previous study, the lower rate of electron–hole recombination was suggested to have a lower PL intensity; thus, it will generate a rapid photodegradation of organic pollutants [53]. However, the findings of this study do not support the previous research. In our case, the NP_seq_-TiO_2_ photocatalyst produced a higher PL signal as compared to the unmodified TiO_2_ and PN_rev_-TiO_2_ photocatalysts. This was because of the presence of N in the NP_seq_-TiO_2_ photocatalyst that was confirmed via EDX and CHNS analysis, capable of forming a new electronic state band, thus increasing the formation of surface oxygen vacancy defects. The excitation electron at the conduction band will go down to the sub-band instead of recombining with the holes at the valence band. Therefore, the reduction process of the NP_seq_-TiO_2_ photocatalyst was taken at the sub-band level. Thus, a higher intensity of the PL signal was produced as the rate of the electron performing a reduction process at the sub-band increased. Moreover, the change in optical absorption ability and the formation of N species and defects could be performed using EPR analysis [54]. Zhang et al. [55] reported that the EPR signal was obtained at g = 2.002, which was attributed to an oxygen vacancy with one electron in the N-doped TiO2 sample. The oxygen vacancy defects attributed to the visible-light absorption will lead to a stability of active nitrogen species. Strong evidence of the NP_seq_-TiO_2_ photocatalyst was found where the wavelength was shifted toward higher wavelengths; hence, it was activated under visible-light energy. Moreover, the particle size of the photocatalyst also plays a major role in producing a stronger PL signal [56]. This is why the NP_seq_-TiO_2_ photocatalyst exhibited a higher PL signal, because it has a smallest particle size and higher specific surface area as compared with other photocatalysts. Furthermore, the PN_rev_-TiO_2_ photocatalyst has a lower recombination rate of electron–hole pairs, which produces an almost flat graph line of the PL spectrum. It can be related to the functionality of the Pt as an electron trapper where it captures the excited electron at the conduction band [57]. Nonetheless, the PN_rev_-TiO_2_ photocatalyst decreased the photocatalytic activity as compared with the NP_seq_-TiO_2_ photocatalyst. This was owed to the larger bandgap energy of the PN_rev_-TiO_2_ photocatalyst, 3.1 eV, which could only be activated under UV light.

### 3.6. Photocatalytic Study

It is crucial to study the mode preparation of the N, Pt co-doped TiO_2_ photocatalyst whereby it gives a significant finding on the percentage of N that was incorporated with TiO_2_. Hence, the photodegradation of the methylene blue (MB) dye was investigated by 60 min of light irradiation for evaluation of the effectiveness of the synthesized process for the N, Pt co-doped TiO_2_ photocatalyst via sequential and vice-versa mode preparations. The photocatalytic activity for all prepared photocatalysts obeyed the Langmuir–Hinshelwood pseudo-first-order rate constant kinetic model where the R^2^ value was obtained in range of 0.9839–0.9986.

The adsorption study was performed in a dark state for 30 min in order to achieve an equilibrium condition before starting the photoactivity experiment with the presence of visible light. Figure 9a shows the percentage remaining of blank, TiO_2_, NP_seq_-TiO_2_, and PN_rev_-TiO_2_ photocatalysts for the photodegradation of 12 mg L^−1^ MB dye irradiated with a 55-Watt fluorescent lamp for 60 min. The control experiment was conducted without the presence of a photocatalyst and showed a slight decrement under visible-light irradiation. This indicates that the photocatalytic process was the main role in the photodegradation of MB dye. It can be seen that the NP_seq_-TiO_2_ sample showed a rapid photodegradation of MB dye and it tended to decolorize within 30 min of light irradiation compared to unmodified TiO_2_ and PN_rev_-TiO_2_, which required 60 min of light irradiation for a complete decolorization. As shown in Figure 9b, the NP_seq_-TiO_2_ photocatalyst gave a higher percentage of MB dye removal of about 95% followed by the PN_rev_-TiO_2_ photocatalyst and unmodified TiO_2_, 90% and 80% MB dye removal, respectively. These are comparable with the rate constant, k, obtained for the NP_seq_-TiO_2_ photocatalyst, which gave the highest photocatalytic activity, ca., 0.1236 min^−1^. The increment in photoactivity of the NP_seq_-TiO_2_ photocatalyst was about 2.0 and 1.5 times higher as compared to the unmodified TiO_2_ and PN_rev_-TiO_2_ photocatalysts, where the rate constants, k, were, ca., 0.0645 min^−1^ and 0.0777 min^−1^, respectively (see Figure 9c).

This was due to the presence of N in the NP_seq_-TiO_2_ photocatalyst linked to the formation of oxygen vacancies and was confirmed by PL spectra. Zhang et al. [58] explained the nitrogen-oxygen bond forming the oxygen defect. In addition, the N decreased the energy bandgap where the absorption edge had a redshift toward the visible-light region, and it was due to the presence of an oxygen vacancy capable of forming specific deep donor levels [59]. Natarajan et al. [60] mentioned that the N dopant is capable of changing the surface-electronic properties, which will improve the photocatalytic activity. Apart from that, the NP_seq_-TiO_2_ photocatalyst had a higher specific surface area and pore volume, which were, ca., 57 m^2^ g^−1^ and 0.61 cm^3^ g^−1^, respectively. Wang et al. [61] explained that the higher specific surface area gives a higher chance for the organic pollutant to be adsorbed onto the surface of the synthesized photocatalyst, while the higher pore volume results in the more rapid diffusion of the inorganic product during the photocatalysis process.

The ICP-OES analysis confirmed that the optimal weight percentage of Pt in the NP_seq_-TiO_2_ and PN_rev_-TiO_2_ photocatalysts was, ca., 2.1 wt.% and 2.2 wt.%, respectively (see Table 2). Thus, the Pt dopant did not have a significant effect by the different preparation of NP_seq_-TiO_2_ and PN_rev_-TiO_2_ photocatalysts. The Pt played its role as electron scavenger, as can obviously be seen in Figure 8c where the PN_rev_-TiO_2_ photocatalyst had the lowest PL spectrum as compared with TiO_2_ and NP_seq_-TiO_2_ photocatalysts. As reported by Mezni et al. [62], the noble metal dopant is attributed to the enhancement of electron lifetime. Nevertheless, the PN_rev_-TiO_2_ photocatalyst gave a lower photoactivity due to the absence of N in the prepared sample, while the higher photocatalytic performance of the NP_seq_-TiO_2_ photocatalyst was caused by the synergistic effect between Pt and N in the prepared sample. Consequently, the different sample preparation pathways for the N, Pt-TiO_2_ photocatalyst had an influence on the N content, which would then influence the photodegradation of the MB dye.

### 3.7. Chemical Oxygen Demand Analysis (COD)

The photocatalytic efficiency of the synthesized sample was evaluated by determining the chemical oxygen demand analysis (COD). It was undertaken to determine the total oxygen content in the sample where the oxygen had a tendency for oxidation of the organic pollutant by forming CO_2_ and water [63]. Thus, it detects the total organic compound in the sample that has not been decomposed. In this research, the COD analysis was conducted for 1 to 7 h of light irradiation using TiO_2_, NP_seq_-TiO_2_, and PN_rev_-TiO_2_ photocatalysts. The COD analysis was taken for 7th cycles of light irradiation for each photocatalyst. Figure 9d shows the trend of the COD value versus time of light irradiation using a 55-Watt fluorescent lamp for the TiO_2_, NP_seq_-TiO_2_, and PN_rev_-TiO_2_ photocatalysts on the photodegradation of MB dye. It was observed that the COD value for TiO_2_, NP_seq_-TiO_2_, and PN_rev_-TiO_2_ photocatalysts decreased as the light irradiation increased. The NP_seq_-TiO_2_ photocatalyst gave the lowest COD value of about 20 mg/L within an hour of light irradiation as compared to the PN_rev_-TiO_2_ and TiO_2_ photocatalysts, which produced higher COD values of about 28 and 30 mg/L, respectively. However, the PN_rev_-TiO_2_ photocatalyst produced a slower mineralization process as compared to the TiO_2_. This might be due to the decrement in specific surface area and pore volume where it affected the photocatalytic process (see Table 1). The NP_seq_-TiO_2_ photocatalyst achieved a complete mineralization after 7 h of light irradiation, followed by PN_rev_-TiO_2_ and TiO_2_ photocatalysts, 5 and 10 mg/L, respectively. Thus, it verified the higher photocatalytic activity of the NP_seq_-TiO_2_ photocatalyst, ca., 0.128 min^−1^, due to the lower O_2_ content during the photodegradation of MB dye, which gave a complete degradation of MB dye. This proves the presence of N in the NP_seq_-TiO_2_ photocatalyst that plays a significant role in increasing the photocatalytic activity about 2 times higher as compared to the PN_rev_-TiO_2_ and TiO_2_ photocatalysts.

### 3.8. Mechanism Reaction

Figure 10 shows the proposed mechanism reaction for the (a) NP_seq_-TiO_2_ photocatalyst and (b) PN_rev_-TiO_2_ photocatalyst. The proposed mechanism reaction is intended to further explain the reason behind the mode preparation that affects the photocatalytic degradation of MB dye. The significant enhancement of photoactivity was seen for the sequential method of the NP_seq_-TiO_2_ photocatalyst. In the sequential method, the calcination process took place for the preparation of the N-TiO_2_ photocatalyst. At this stage, the N was incorporated with the TiO_2_ structure and potentially formed a strong Ti-O-N and/or Ti-N bonding. The presence of strong N bonding introduced a new sub-band located above the TiO_2_ valence band. Thus, it shifted the absorption edge toward the visible-light region. In addition, the N promoted the formation of oxygen vacancies, which could easily capture or attach with the photoinduced electrons. Subsequently, the Pt was introduced into the N-TiO_2_ photocatalyst through the photodeposition process, carried out by mixing the Pt precursor with the N-TiO_2_ photocatalyst to synthesize the NP_seq_-TiO_2_ photocatalyst. Wang et al. [64] stated that the formation of the Schottky barrier at the surface of the modified photocatalyst was due to the interaction between the Pt and N-TiO_2_ photocatalyst, and it caused the bending of conduction and valence bands toward the N–TiO_2_ interface. Therefore, the excitation electron from the valence to conduction band will be captured by the Pt dopant to undergo the reduction process, forming the superoxide radicals. These radicals will determine the degradation of MB dye.

Meanwhile, reverse preparation was applied to synthesize the PN_rev_-TiO_2_ photocatalyst. In this route, the photodeposition process was conducted to produce the Pt-TiO_2_ photocatalyst followed by the calcination process with a mixture of urea forming the PN_rev_-TiO_2_ photocatalyst. The PN_rev_-TiO_2_ photocatalyst was found to have a lower photocatalytic activity than the NP_seq_-TiO_2_ photocatalyst. According to Cybula et al. [65], a higher calcination temperature will increase the crystallite size and reduce the surface area of the photocatalyst. This statement is relatable as the PN_rev_-TiO_2_ photocatalyst had a lower specific surface area and pore volume and the highest crystallite size, ca., 40.96 nm. Thus, the changes in the crystallite structure as well as surface properties had an influence on the photoactivity of the PN_rev_-TiO_2_ photocatalyst. In addition, the absence of N in the synthesized PN_rev_-TiO_2_ photocatalyst was detected through EDX and CHNS analysis. This explained that the N element was not embedded onto the Pt-TiO_2_ structure but only attached surrounding the photocatalyst, which easily detached when the washing and photodegradation processes were applied. Therefore, the optical properties for the PN_rev_-TiO_2_ photocatalyst did not change, as no presence of N in the PN_rev_-TiO_2_ photocatalyst was observed. It was confirmed by the UV-Vis/DRS analysis where the bandgap energy was, ca., 3.1 eV, respectively. Thus, the method of preparation for the modification of N and Pt with commercially available TiO_2_ is critical to study to improve the photocatalytic activity of the TiO_2_ photocatalyst. 

## 4. Conclusions

The results of this study indicate that the sequence of two different processes on the synthesis of the modification of TiO_2_ with N and Pt dopants is crucial to investigate, whereby it gives a different photoactivity performance. Therefore, the NP_seq_-TiO_2_ photocatalyst was successfully prepared via the sequential mode and was observed as having a higher photocatalytic activity. This was due to the presence of N playing a major role in enhancing the photoactivity by increasing the formation of OH bonding and defects in the TiO_2_ structure by forming oxygen vacancies, and also reducing the bandgap energy from 3.2 eV to 2.9 eV; thus, the optical properties shifted toward the visible-light region. In addition, the higher surface area increased the photoactivity of the NP_seq_-TiO_2_ photocatalyst due to the rapid adsorption process of the MB dye on the NP_seq_-TiO_2_ photocatalyst’s surface. The sequential mode preparation of the N, Pt–TiO_2_ photocatalyst was successfully produced and a great understanding of the reaction mechanism was attained. This research will act as a reference for future studies on the route preparation in combining N and Pt dopants for modification of metal oxide semiconductors. The synthesized NP_seq_-TiO_2_ photocatalyst was limited to the degradation of organic pollutants in the wastewater treatment application.

## Figures and Tables

**Figure 1 nanomaterials-12-03998-f001:**
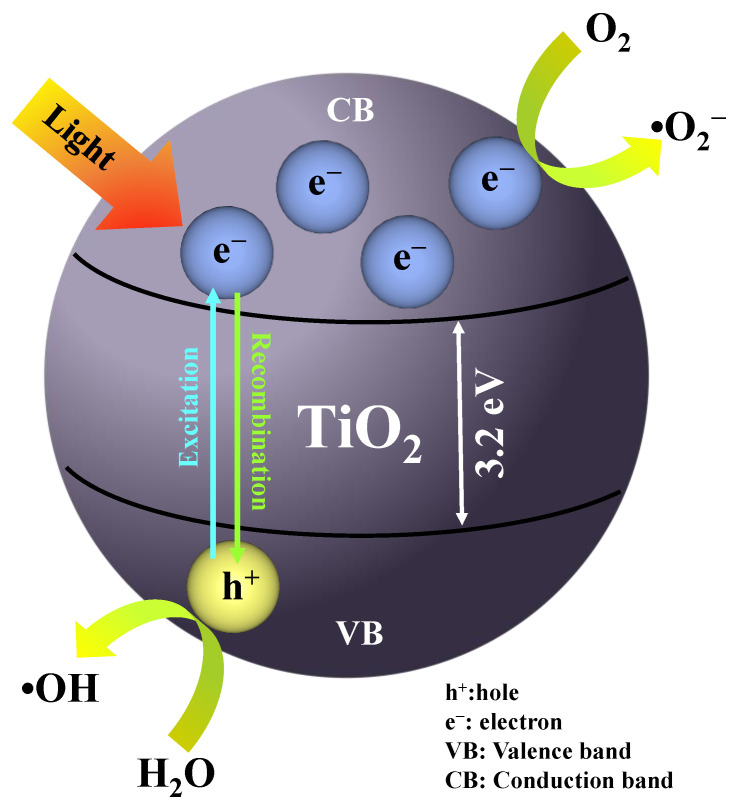
The excitation and recombination process of electron-hole pair.

**Figure 2 nanomaterials-12-03998-f002:**
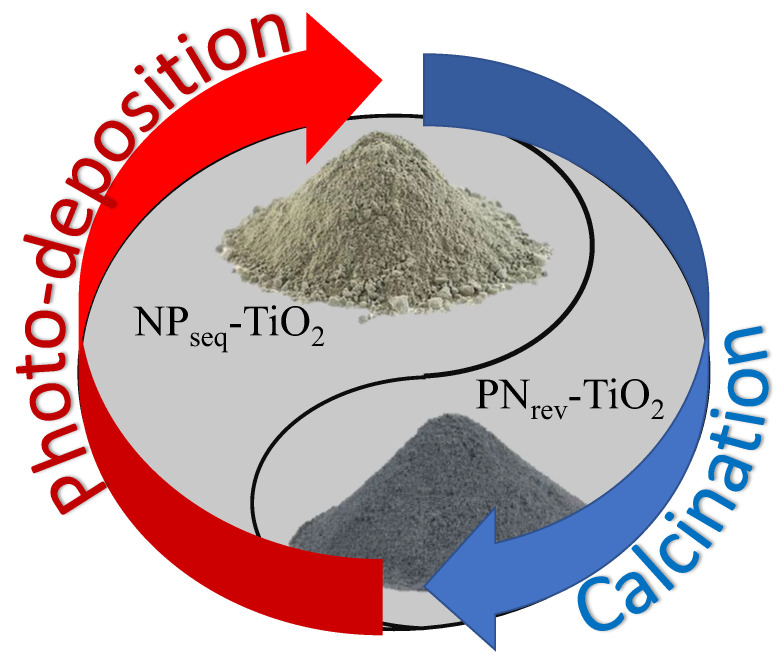
The mode preparation for NP_seq_-TiO_2_ and PN_rev_-TiO_2_ photocatalysts.

**Figure 3 nanomaterials-12-03998-f003:**
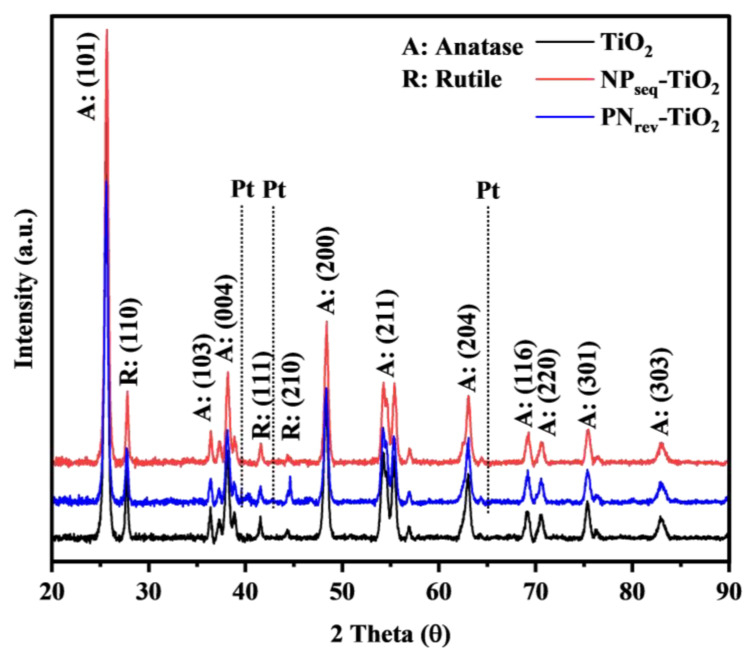
XRD analysis for TiO_2_, NP_seq_-TiO_2_, and PN_rev_-TiO_2_ photocatalysts.

**Figure 4 nanomaterials-12-03998-f004:**
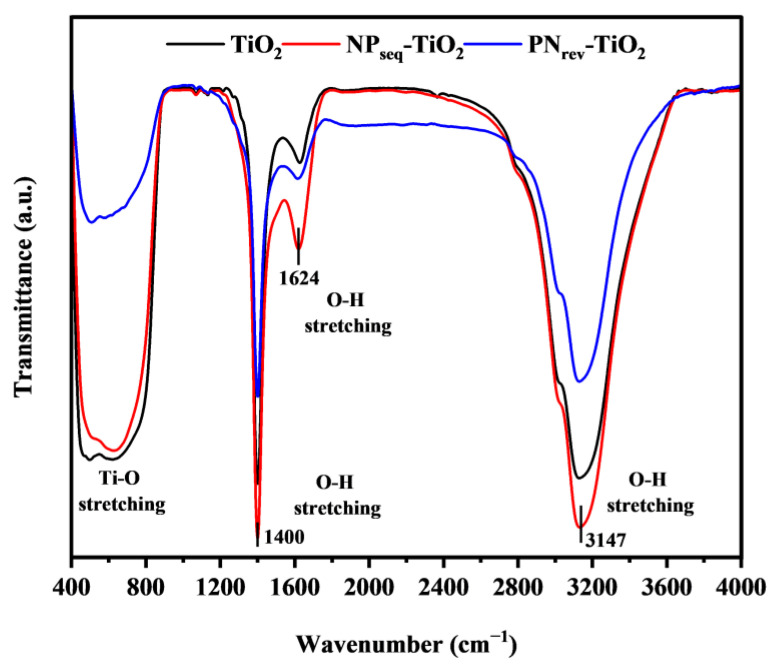
The FTIR spectrum for TiO_2_, NP_seq_-TiO_2_, and PN_rev_-TiO_2_ photocatalysts.

**Figure 5 nanomaterials-12-03998-f005:**
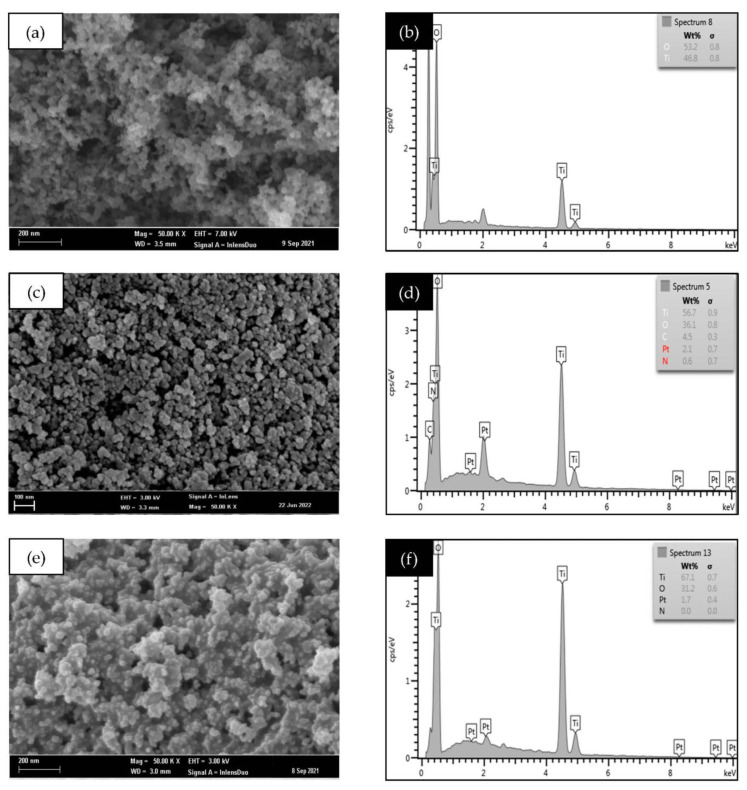
The morphology and EDX analysis for (**a**,**b**) TiO_2_, (**c**,**d**) NP_seq_-TiO_2_, and (**e**,**f**) PN_rev_-TiO_2_ photocatalysts.

**Figure 6 nanomaterials-12-03998-f006:**
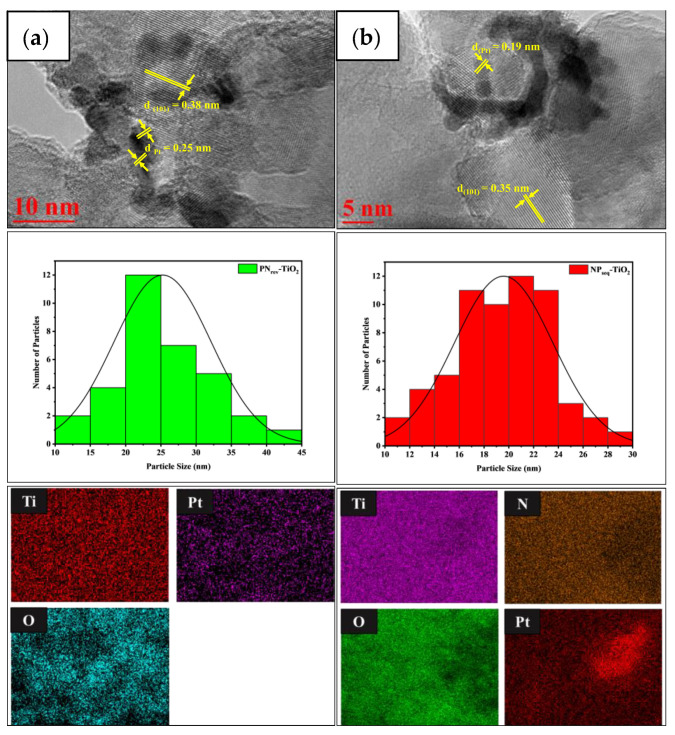
The HR-TEM image, average particle size, and mapping analysis for (**a**) NP_seq_-TiO_2_ and (**b**) PN_rev_-TiO_2_ photocatalysts.

**Figure 7 nanomaterials-12-03998-f007:**
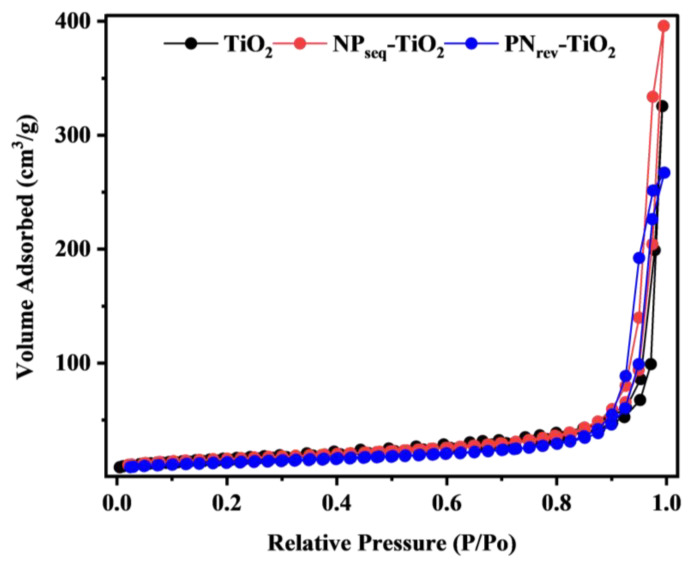
The nitrogen adsorption–desorption isotherm for TiO_2_, NP_seq_-TiO_2_, and PN_rev_-TiO_2_ photocatalysts.

**Figure 8 nanomaterials-12-03998-f008:**
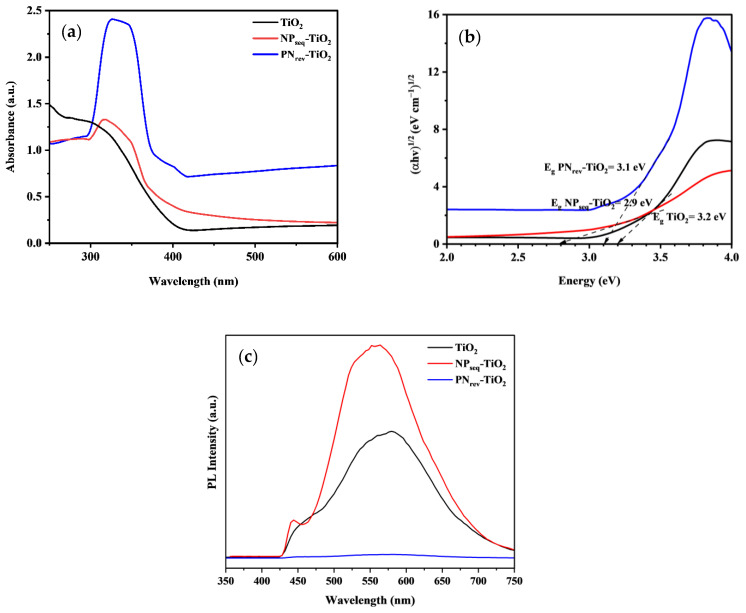
The UV-Vis/DRS analysis for TiO_2_, NP_seq_-TiO_2_, and PN_rev_-TiO_2_ photocatalysts: (**a**) absorbance spectra, (**b**) Tauc’s plot, and (**c**) PL spectra for TiO_2_, NP_seq_-TiO_2_, and PN_rev_-TiO_2_ photocatalysts.

**Figure 9 nanomaterials-12-03998-f009:**
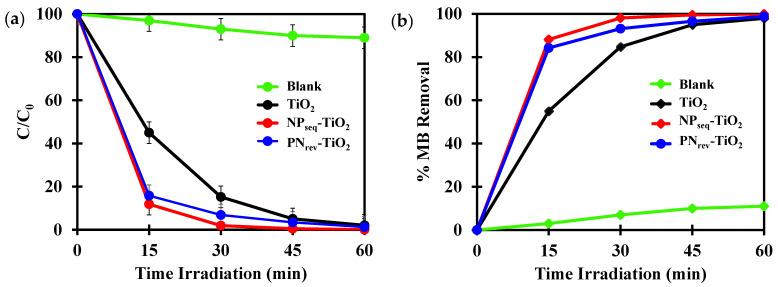
(**a**) The percentage remaining of MB dye; (**b**) the percentage removal of MB dye; (**c**) the rate constant, k; (**d**) the chemical oxygen demand for TiO_2_, NP_seq_-TiO_2_, and PN_rev_-TiO_2_ photocatalysts.

**Figure 10 nanomaterials-12-03998-f010:**
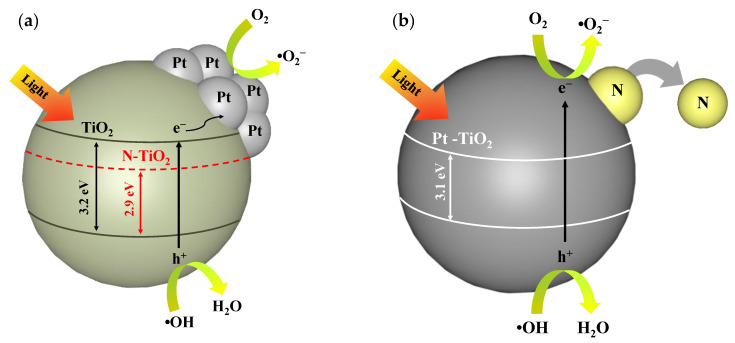
The mechanism reaction for (**a**) NP_seq_-TiO_2_ photocatalyst and (**b**) PN_rev_-TiO_2_ photocatalysts.

**Table 1 nanomaterials-12-03998-t001:** The structural properties of TiO_2_, NP_seq_-TiO_2_, and PN_rev_-TiO_2_ photocatalysts.

Sample	CI (%) ^a^	D_XRD_ (nm) ^b^	S_BET_ (m^2^ g^−1^) ^c^	V (cm^3^ g^−1^) ^d^	L (nm) ^e^	E_g_ (eV) ^f^
TiO_2_	81.33	35.43	54	0.50	37.45	3.20
NP_seq_-TiO_2_	80.81	32.86	57	0.61	43.43	2.90
PN_rev_-TiO_2_	79.99	40.96	25	0.41	67.3	3.10

^a^ Crystallinity index. ^b^ Crystallite size. ^c^ Specific surface area. ^d^ Pore volume. ^e^ Average pore diameter. ^f^ Bandgap energy.

**Table 2 nanomaterials-12-03998-t002:** The sample preparation and pseudo-first-order rate constant, k (min^−1^), for photocatalytic degradation of 12 mg L^−1^ of MB dye for TiO_2_, NP_seq_-TiO_2_, and PN_rev_-TiO_2_ photocatalysts irradiated with 55-Watt fluorescent lamp for 60 min.

Sample	Sample Preparation							
	CH₄N₂O (g)	H_2_PtCl_6_ (wt. %)	N ^a^ (wt. %)	N ^b^ (wt. %)	Pt ^a^ (wt. %)	Pt ^c^ (wt. %)	k (min^−1^)	R^2^
TiO_2_	-	-	-	-	-	-	0.0645	0.9986
NP_seq_-TiO_2_	1.0	2.0	0.60	1.58	2.10	2.10	0.1236	0.9980
PN_rev_-TiO_2_	1.0	2.0	-	-	1.70	2.20	0.0777	0.9839

^a^ Detected by EDX analysis. ^b^ Detected by CHNS-elemental analyzer. ^c^ Detected by ICP-OES analysis.

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
