# Peer review of "The Role of Nitrogen-Doped TiO2 Supported by Platinum Catalyst Synthesized via Various Mode Preparations for Photocatalytic Enhancement"

_nanomaterials, 2022, doi:10.3390/nano12223998_

Round 1
Reviewer 1 Report
The article entitled “The Role of Nitrogen doped TiO2 Supported by Platinum Catalyst Synthesized via Various Mode Preparation for Photocatalytic Enhancement”was reviewed. This work and the results were interesting. In this manuscript, NPseq-TiO2 and PNrev-TiO2 were synthesized. This manuscript can be accepted for publishing “Nanomatereals” but I have some major remarks before it can be published.
1.If possible, more direct verifications should be provided to identify the surface structures, such as HRTEM and XPS.
2.It is suggested to add a blank contrast experiment without catalyst in Fig.8.
3. Does the adsorption equilibrium process need to be performed before photocatalytic degradation of MB? It is recommended to add the corresponding data in Fig. 8.
4. If possible, more verifications should be provided to identify the promoted charge separation, such as time-resolved transient absorption spectra analysis and transient PL spectrum.
5. The N2 adsorption/desorption isotherms and pore size distribution curves should be provided.
6. The active oxidation species for more samples should be carefully investigated by the different methods, such as ESR or EPR.
Reviewer 2 Report
Referee report on “The Role of Nitrogen doped TiO2 Supported by Platinum Catalyst Synthesized via Various Mode Preparation for Photocatalytic Enhancement" by N.S. Natar et al.
Although this topic is of some interest, this manuscript in its present form cannot be recommended for publication and requires at least some improvement.
1. Introduction. 1 paragraph. Sentence “Semiconductor has a band gap energy (Eg) which was defined as the energy difference of the valence band (VB) and conduction band (CB).” This is not entirely correct, as it applies to the insulator and by the way, CeO2 is a typical insulator.
2. Although the introduction is well written, several points are either not disclosed or not completely accurate. In particular, nothing is said about the morphology of TiO2-based materials and how morphology affects the functional properties. A lot of recent studies have been published in MDPI journals. It is known that TiO2 can be in different crystalline modifications, and the nanostructure is also very diverse, both in structure and size. Few of them are below:
Serga, V.; et al Crystals 2021, 11, 431. https://doi.org/10.3390/cryst11040431
Lin, Y.-P.; et al. Nanomaterials 2021, 11, 2900. https://doi.org/10.3390/nano11112900
Tsebriienko, T.; et al. Crystals 2021, 11, 794. https://doi.org/10.3390/cryst11070794
3. Table 1. Can the data really be given with such precision? Error bars must be included !
4. Fig.5. The quality of the all figures needs serious improvement, as many details are indistinguishable.
5. Figure 6. Definition Eg, looks very arbitrary. The unambiguity of the obtained values requires a separate justification.
6. Figure 7. The wavelength of the excitation light is not specified.
7. Figure 8. The exposure time is not a specific value because it depends on the intensity and wavelength of the excitation source. It is much better to use the values of photons/cm2.
8. References are not in MDPI format and should be edited
Reviewer 3 Report
-
This is a very meaningful work.
N and Pt co-dopant TiO2 photocatalyst was prepared under different mode preparations. The results show that the photoactivity under methylene blue (MB) dye degradation of NPseq -TiO2 photocatalyst is 2 and 1.5 times as faster as compared to the unmodified TiO2. The explanations on machenism are acceptable.
shortcomings: There are somthing wrong about numbering in fig.5
Reviewer 4 Report
In this manuscript, The Role of Nitrogen-doped TiO2 Supported by Platinum Catalyst Synthesized via Various Mode Preparation for Photocatalytic Enhancement. However, there are some minor issues that need to be rectified before the acceptance. Abstract: In the abstract, the results of the research must be briefly described.
1. Abstract requires more technical achievements from the proposed work to highlight the novelty of the work.
2. Keywords should not identical to title words. Should revise it.
3. The authors should clearly explain the innovation, research gap, market gap, market demand, and importance of their work in the manuscript's introduction. They should justify the value of the work and compare their work with previously similar published papers.
4. The authors did not explain the limitations. Should I assume that there are no limitations? It would be nice if they said the future perspectives and their limitations in conclusion, which can attract more readers.
Reviewer 5 Report
In this paper, the authors synthesized two different Pt-N/TiO2 by changing the order of photo-deposition and calcination treatments. Further photocatalytic degradation experiments showed that the prepared photocatalysts were able to degrade the target pollutant MB under visible light irradiation. I would like to see published in Nanomaterials once the following issues are addressed.
1. The authors should use STEM to measure both catalyst samples and combine the analysis with STEM-EDX, especially for Pt nanoparticles. I believe that the Pt nanoparticles formed large-sized particles during calcination (also evidenced by the authors in Figure 5g), thus leading to different photocatalytic activities.
2. The analysis of the deconvolution of the FTIR spectrum (Figure 4) is inaccurate. The peaks chosen by the authors do not have a clear physical meaning (there are no clear criteria for the number, position, FWHM and other parameters of the peaks). It is suggested that the authors remove this section and add XPS data (including Ti, O, N, Pt peaks)
3. The specific surface area measurement and adsorption MB experiment of catalyst should be supplemented
4. The authors labeled “N molecule leaching out” in Figure 9, which needs further explanation. Why does this phenomenon not occur for NPseq-TiO2 sample? And N molecule is not leaching out.
5. The authors did not provide any catalyst cycling data. And the authors needed to measure the physical and chemical properties of the catalyst before and after the cycle.
6. Additional free radical capture experiments and ESR data of reactive radicals should be indicated for the photocatalytic mechanism study.
7. Authors should double-check their manuscripts before submitting a revision, there are so many errors. E.g. no figure 5h, etc, also the scales in Figure 5a,b,c need to be clearly marked.
8. Several additional articles on TiO2 photocatalysis modified by other strategies are suggested. (e.g., 10.1016/j.apcatb.2016.03.021, 10.3390/nano9030391, 10.1016/j.apcatb.2017.03.077).
Round 2
Reviewer 1 Report
no
Reviewer 2 Report
The authors have constructively improved their
Original manuscript, which now can be recommended for publication.
Reviewer 5 Report
My comments were addressed satisfactorily. The manuscript is much improved and I'd be happy to see it published in nanomaterials.